# Small Physical Cross-Linker Facilitates Hyaluronan Hydrogels

**DOI:** 10.3390/molecules25184166

**Published:** 2020-09-11

**Authors:** Saliha Erikci, Patricia Mundinger, Heike Boehm

**Affiliations:** 1Department of Cellular Biophysics, Max Planck Institute for Medical Research, 69120 Heidelberg, Germany; saliha.erikci@mr.mpg.de (S.E.); Patricia.Mundinger@sanofi.com (P.M.); 2Physical Chemistry, Heidelberg University, 69120 Heidelberg, Germany

**Keywords:** hydrogel, hyaluronic acid, physical- and chemical cross-link, biocompatibility, cell encapsulation, tissue engineering

## Abstract

In this study, we demonstrate that small charged molecules (NH_4_^+^, GluA^+^, dHA^+^) can form physical cross-links between hyaluronan chains, facilitating polymerization reactions between synthetically introduced thiol groups (HA-DTPH). These hybrid hydrogels can be obtained under physiological conditions ideally suited for 3D cell culture systems. The type and concentration of a physical crosslinker can be adjusted to precisely tune mechanical properties as well as degradability of the desired hydrogel system. We analyze the influence of hydrogen bond formation, concentration and additional ionic interactions on the polymerization reaction of HA-DTPH hydrogels and characterize the resulting hydrogels in regard to mechanical and biocompatibility aspects.

## 1. Introduction

Hyaluronic acid (HA), a major component of the extracellular matrix (ECM), is a non-sulfated glycosaminoglycan (GAG) consisting of repeating disaccharide units (1,4-D-glucuronic acid and−1,3-N-acetyl-D-glucosamine) [1]. HA exhibits excellent biocompatibility and biodegradability and has many important biological functions, such as stabilizing and organizing the ECM, regulating cell adhesion and motility, and mediating cell proliferation and differentiation [2]. As a consequence, HA and its derivatives have become widely used in clinical medicine. In particular, HA-based hydrogels prepared from chemically modified HA have significant potential regarding tissue regeneration, drug delivery and cell encapsulation [3].

To synthesize chemically modified HA three different modification sites on the HA structure are suitable. Modification can proceed at the carboxyl group on the glucuronic acid moiety, at the hydroxyl group and amine group (after deacetylation) on the N-acetyl-D-glucosamine moiety. Different cross-linking methods can be chosen according to the functionalization of the HA molecule.

A widely used example of an HA modification is the functionalization on the carboxyl groups of the glucuronic acid moiety with thiol groups (HA-DTPH) enabling disulfide bond formation and leading to hydrogels via oxidation with hydrogen peroxide [4]. Another cross-linking method of thiol modified HA is based on Michael thiol addition with different acrylate groups [5]. The mechanical properties of hydrogels based on thiol functionalized HA can be regulated through the molecular weight of HA, the percentage of modification and the ratio between thiol functionalized HA and used acrylate molecule. Based on the Michael thiol addition, the commercially available Extracel, ^TM^-HPG has been developed by Glykosan BioSystems Inc. (Salt Lake City, UT, USA) offering a chemically cross-linked HA-based hydrogel. It is based on thiol-modified HA and thiol-modified gelatin, which are cross-linked by poly (ethylene glycol) diacrylate. It is employed in stem cell cultivation, tissue engineering and cell therapy.

Furthermore, hyaluronan can be modified at the hydroxyl groups, the reducing end and the acrylamide group to be functionalized with azides for cycloaddition reactions, methacrylate groups, aldehydes or other cross-linking groups suitable for aqueous solution reactions [6]. Alternatively, HA-based hydrogels can be manufactured by combining HA with a polycation through physical cross-linking. For this, chitosan, which occurs naturally as a polycation, is for example combined with HA and chitosan-HA fibers are formed via self-assembling [7].

In general, cross-linked polymers with a high affinity for water can form water-swollen cross-linked hydrogels. These hydrogels can generally be formed via different methods of cross-linking, such as physical and chemical cross-linking. In chemically cross-linked networks, the polymer backbones are linked via a covalent bond. In contrast, physically cross-linked networks are held together by physical interactions such as ionic interactions, hydrogen bonding or hydrophobic interactions. The chemical cross-linking procedure often requires cross-linking agents, which have to be removed afterwards by different washing steps. Not only impurities left from the preparation process, but also non-reacted monomer, initiators, as well as unwanted side products have to be considered. Aiming at a clinical use, such as tissue engineering or drug release, hydrogel synthesis requires biocompatible hydrogel development avoiding cytotoxic coupling agents. For the investigation of signaling pathways or an in vivo phenotype, the tissues and cells require an in vivo inspired 3D environment. The process by which 3D environments, such as hydrogels, are formed must also be biocompatible due to the fact that cells are present during the gelation process [8]. An additional challenge regarding 3D cell environments and tissue engineering is to mimic the mechanical and biological properties of the desired ECM environment and to, furthermore, be able to control the mechanical and biological properties and their adaptability [9].

Here we introduce physical cross-links to facilitate chemical cross-links of functionalized HA under biocompatible polymerization conditions ideally suited for cell-encapsulation. Additionally, the mechanical properties of the resulting hydrogel can be tuned based on the type and concentration of incorporated physical cross-linkers.

## 2. Results

### 2.1. Hydrogel Synthesis Induced by Physical Cross-Links

Combining thiol modified HA with an ionic cross-linker leads to hydrogel formation under argon atmosphere at room temperature without the need of hydrogen peroxide. Three different ionic cross-linkers were chosen to analyze the effect of physical cross-links on the overall HA-hydrogel properties: (1) an HA-disaccharide unit (dHA^+^) based ionic cross-linker, (2) charged glucosamine (GluA^+^) and (3) ammonium chloride (NH_4_^+^). The used ionic cross-linkers were also chosen because of their increasing capacity to form hydrogen bonds from NH_4_^+^ to GluA^+^ and dHA^+^.

The HA inspired ionic cross-linker dHA^+^ was synthesized by acidic degradation and further deacetylation via incubation with H_2_SO_4_ for 6 h at 100 °C and was purified subsequently by applying an amberlite column. The ninhydrine positive fractions are collected and are concentrated under reduced pressure to get the desired product.

Successful hydrogel formation was determined by utilizing an inverted tube test. Dissolved HA-DTPH (3% *w*/*v*) in PBS without any cross-linker at a pH range of 7, 5–8, 5 showed no increase in viscosity, neither under oxygen nor argon atmosphere for 3 days. The solution was still flowing after 3 days of incubation in an inverted test tube assay (Figure 1e). Subsequently overlaying this solution with 0.3% H_2_O_2_ after 3 d incubation within oxygen atmosphere led to hydrogel formation within 1 h (HA-DTPH-Ox.). This hydrogel system is assumed to be based only on chemical cross-links. In the presence of NH_4_^+^ (Figure 1b), GluA^+^ (Figure 1c) and dHA^+^ (Figure 1d), with a crosslinker ratio of 1.0 to the free remaining negative groups on HA-DTPH, dissolved HA-DTPH (3% *w*/*v*) showed stable hydrogel formation after 24 h under argon atmosphere in inverted test tube assays (Figure 1).

### 2.2. Ionic Cross-Linker Enhances Disulphide Bond Formation

Adding either dHA^+^, GluA^+^ or NH_4_^+^ to HA-DTPH polymerization mixture under argon atmosphere significantly facilitated the disulfide bond formation. Hydrogels based on HA-DTPH with varying degrees of thiolation and one of the respective ionic cross-linkers are prepared with a 1:1 ratio of ionic cross-linker to un-functionalized carboxy groups of the respective HA-DTPH. Subsequently, an Ellman’s assay is performed over time. For HA-DTPH^29%^ without an ionic cross-linker (Figure 2a), the measured free thiol amount decreased from 100% free thiol groups to 36 ± 0.08% within 24 h, whereas for HA-DTPH^29%^-NH_4_^+^, a decrease from 100% to 33 ± 0.03%, for HA-DTPH^29%^-GluA^+^ from 100% to 10% ± 0.02 and for HA-DTPH^29%^-dHA^+^ from 100% to 7 ± 0.02% was observed. The same trend is seen for HA-DTPH^42%^ (Figure 2b) and HA-DTPH^58%^ (Figure 2c) with and without the cross-linker, respectively.

### 2.3. Young’s Modulus Dependency on Chemical and Physical Cross-Links

The Young’s modulus of the HA-DTPH-Cl^+^ hydrogels was shown to be dependent on the amount of chemical cross-link, varied by modulating the thiolation degree of HA, as well as the amount of physical cross-link varied by using three different ionic cross-linkers and different concentration of the ionic cross-linkers within the hydrogel network. The Young’s modulus of the established HA-DTPH-Cl^+^ hydrogels were measured in a fully swollen state and compared to the HA-DTPH-Ox. hydrogels. The measurements revealed a significant difference between HA-DTPH-Ox. and HA-DTPH-Cl^+^ hydrogels. HA-DTPH^29%^-Ox. hydrogels, after being swollen in H_2_O, have the lowest Young’s modulus, which increases with increasing thiolation degree. As for HA-DTPH-Cl^+^ hydrogels, the Young’s modulus increases with increasing capability to form hydrogen bonds of the used ionic cross-linker. A common trend for both HA-DTPH-Ox. and HA-DTPH-Cl^+^ is the increased Young’s modulus with increasing thiolation degree (Figure 3).

The tunability of the stiffness was also proven to be achievable by varying the concentration of the used ionic cross-linker dHA^+^ at a constant degree of thiolation of HA-DTPH. Increasing the ionic cross-linker dHA^+^ concentration resulted in an increased Young’s modulus (Table 1).

### 2.4. Swelling Ratio

The swelling ratio of HA-DTPH-Cl^+^ hydrogels and HA-DTPH-Ox. hydrogels significantly depends on the degree of thiolation, the type of the cross-linker and the ionic strength of the solution. HA-DTPH-Ox. show the lowest swelling ratios compared to HA-DTPH-Cl^+^ hydrogels. For HA-DTPH-Cl^+^, a clear trend of the swelling behavior is observed. Swelling ratio is increasing from dHA^+^ and GluA^+^ to NH_4_^+^. A common trend for each hydrogel system is the decreased swelling ratio with increasing ionic concentration. This accounts for all ions tested. The second trend is the decreasing swelling ratio from NaCl to CaCl_2_. The lowest swelling ratio was determined for hydrogels incubated with CaCl_2_. The third feature is the decreased swelling ratio with increasing thiolation degree (Figure 4) (see Figure A1 of Appendix A).

### 2.5. Cross-Linking Dependent Enzymatic Degradation

The degradation of HA-DTPH-Ox. hydrogels was significantly faster than the degradation of HA-DTPH-dHA^+^ hydrogels. In general, the degradability of hydrogels is an important property of a hydrogel system, especially for the cultivation of encapsulated cells. For example, Khetan et al. 2013 [10], have been able to show that encapsulated stem cell differentiation was induced by cellular traction modulated through the degradability of the hydrogel.

In vitro, degradation rates of the HA-DTPH-dHA^+^ hydrogels and HA-DTPH-Ox. hydrogels were found by incubating hydrogels with two different enzymes, namely hyaluronidase and hyaluronate lyase, at concentrations of 100 U/mL in PBS, to ensure complete gel degradation before hyaluronidase IV inactivation [11]. The half-time t_1/2_ of hydrogels defines the time at which the weight of the gel is reduced to half the initial weight and were calculated from the initial linear slope of gel mass vs. time plots. Significant differences were measured in the degradation rates comparing HA-DTPH-Ox. and HA-DTPH-dHA^+^. HA-DTPH-Ox. hydrogels have shorter half-lives than comparable HA-DTPH-dHA^+^ made with the same HA-DTPH (Figure 5). Half-lives for hyaluronate lyase (Figure 5a) are shorter with 1.8 ± 0.3 h for HA-DTPH^29%^-dHA^+^ to 0.3 ± 0.03 for HA-DTPH^29%^-Ox. For hyaluronidase IV (Figure 5b), half-lives range from 2.2 ± 0.3 h for HA-DTPH^29%^-dHA^+^ to 0.5 ± 0.03 h for HA-DTPH^29%^-Ox. hydrogel. No significant change in mass was observed in the negative-control for HA-DTPH-dHA^+^ hydrogels and HA-DTPH-Ox. hydrogels, which were incubated in PBS buffer alone. Long-term stability measurements were performed over six months in PBS and HA-DTPH-dHA^+^ hydrogels were weighed each week, which showed a weight loss of less than 10% after six months (data not shown).

### 2.6. Cell Encapsulation in HA-DTPH^58%^-dHA^+^ Hydrogel

Normal human dermal fibroblasts (NHDF) were successfully embedded inside the HA-DTPH^58%^-dHA^+^ hydrogel and cell viability was proven after 72 h. Further incorporation of RGD ligand, covalently coupled to HA molecules inside the hydrogel, induced cell spreading. Successful tissue engineering requires cell viability of encapsulated cells. The gelation time and biocompatible hydrogel formation of the HA-DTPH-dHA^+^ demonstrated the capacity for in situ cell encapsulation. Viable cells, indicated by green fluorescence upon Calcein AM staining, were evident, whereas fewer than 5% dead cells were observed through ethdidium homodimer staining (Figure 6a). Incorporation of linear RGD within the hydrogel, promoted cell attachment and spreading of the encapsulated NHDF cells. Successful attachment and spreading were proven by DAPI and Phalloidin staining (Figure 6b).

## 3. Discussion

In general, the functionality and biocompatibility of the hydrogel can be regulated by carefully designing the cross-linking chemistry. Combining both chemical and physical cross-linking methods resulted in a novel functionalized HA-based hydrogel containing permanent junctions by the formation of chemical cross-links and transient junctions by the formation of physical cross-links. Each proved to have a high impact on the hydrogel properties. First of all, by using an ionic cross-linker, the polymerization reaction can be carried out under biocompatible conditions and the use of hydrogen peroxide to oxidized free thiol groups can be avoided. The quick formation of disulfide bonds in the presence of esp. strong hydrogen building cross-linkers is most likely enabled through shielding of the negative groups on HA-DTPH by the ionic cross-linkers, as well as through shorter distances between different thiol groups, facilitated by initial physical crosslinks between different polymer chains mediated by the ionic cross-linkers.

The Young’s modulus of hydrogels can restrict their use for certain biomedical applications. Therefore, it is crucial to have a system which is tunable in its stiffness and through this covers a broad range of tissues. Varying the chemical and physical cross-links by varying either the thiolation degree or the ionic cross-linker structure and concentration resulted in a tunable Young’s modulus. Here, an increasing thiolation degree leads to a higher ratio of disulfide bonds, which results in a tighter network. Comparing HA-DTPH-Cl^+^ hydrogels to HA-DTPH-Ox. hydrogels, the influence on the stiffness of the physical cross-link is remarkable. The differences of the Young’s modulus between HA-DTPH-Ox. and HA-DTPH-Cl^+^ hydrogels strengthened the hypothesis of the influence of additionally formed physical cross-links on the mechanical stiffness. Due to the formation of hydrogen bonds and salt bridges, the network tightens and becomes stiffer. The decreasing Young’s modulus from dHA^+^ to GluA^+^ and to NH_4_^+^ also supports the hypothesis that the increasing capability of ionic cross-linkers to from hydrogen bonds leads to a stabilizing and tightening of the network.

In order to determine the uptake of water and the sensitivity to changing ionic strength in dependency of the thiolation degree and the ionic cross-linker, we investigated swelling behavior of the hydrogels in water and at different ionic strengths in water and in the presence of monovalent and divalent ions. Generally, the swelling ratio decreases with (1) increasing capacity of the ionic cross-linker to form hydrogen bonds and with (2) increasing number of thiol bridges at higher degrees of thiolation. This can be attributed to the rising cross-links within the hydrogel network, which results in a decreasing capacity of the hydrogels to take up water. A remarkably surprising result is the swelling ratio of HA-DTPH-Ox. compared to the swelling ratio of HA-DTPH-Cl^+^. Although its Young’s modulus is softer, HA-DTPH-Ox shows the lowest swelling ratio. The cause for these different swelling behaviors of HA-DTPH-Cl^+^ and HA-DTPH-Ox. hydrogel could be ascribed to several different mechanisms. By introducing different ionic cross-linkers with several hydrophilic chemical residues like hydroxyl groups (-OH) in case of dHA^+^ and GluA^+^ the water absorbing properties within the network increases. Another mechanism could be by the so called forward osmosis. In this case water molecules diffuse, following the osmotic pressure, from a solution with lower osmotic pressure to another solution with higher osmotic pressure. By adding charged molecules to the hydrogels, the osmotic pressure inside the hydrogel network increases, and thus the water diffuses into the hydrogel. Such a mechanism of osmotic driving forces during the hydrogel swelling process gives the opportunity to tune the correlation between the stiffness and swelling ratio. Tuning the dependence of hydrogel stiffness and swelling ratio was also described by Cha et al. (2011) [12]. He incorporated a polymer chain with hydrophobic moieties inside of a poly (ethylene glycol) diacrylate (PEGDA) hydrogel and observed a decrease in the Young’s modulus, by tuning the cross-links within the hydrogel, which caused just a minimal increase in the swelling ratio, due to the hydrophobic moieties. The difference of the swelling ratio observed in the presence of Ca^2^ compared to Na^+^ might be explained by their different capability to form complexes with the carboxylate groups. Such a system could also be used in a chelation therapy to treat metal-interactions by selectively binding metal-ions like shown by Polomoscanic et al. (2005) [13]. They established hydrogels containing hydroxamic acid groups as chelators for iron in the gastrointestinal tract. In general, modifying the whole network charge using a charged molecule is a great tool to regulate hydrogel properties regarding swelling ratio and viscoelasticity. This was also demonstrated by Hegger et al., by designing two structurally similar covalent cross-linkers with either a neutral or a positively charged aromatic core. The overall network structure of the hydrogel was changed by ionic interactions when using the positively charged crosslinker, since this reduced the negative charge, further affecting counter ion density. Using these cross-linkers and varying the degree of thiolation, Hegger et al. also observed that increasing the thiolation degree led to a higher Young’s modulus. Moreover, hydrogels linked with the positively charged crosslinker displayed up to two times higher Young’s moduli. These interactions also affected the swelling ratios, which decreased with an increasing thiolation degree and with a reduced negative charge due to the use of the positively charged crosslinker [5].

Testing the enzymatic degradability of the novel hydrogel system compared to the oxidized hydrogel system without additional physical cross-links indicated a higher stability against the enzymatic degradation of the hydrogels reinforced through physical cross-links. In general, the experiments showed that the HA-DTPH-dHA^+^ hydrogel system is still degradable, and thus it fulfills an important property for biomedical application. Adjusting the concentration of dHA^+^ therefore also enables the adjustment of the HA-DTPH-dHA^+^ hydrogel system towards enzymatic degradation, which makes it more flexible and adjustable.

Due to the viability of encapsulated NHDF cells after 72 h, post-encapsulating within the HA-DTPH-dHA^+^ hydrogel proved the biocompatible hydrogel polymerization. Increasing the complexity of the hydrogel system by adding a liner RGD binding motive, we aimed to address integrin-mediated cell attachment and spreading. This provides a hydrogel system suitable for tissue engineering as a 3D inspired in vivo environment.

## 4. Materials and Methods

Research grade hyaluronic acid (average molecular weight 74 kDa) was obtained from Lifecore Biomedical (Chaska, MN, USA). The cross-linkers for hydrogel formation were synthesized adapted from a protocol of Vibert et al. 2009 [14]. The compounds glucosamine hydrochloride and ammonium chloride, hyaluronidase type IV from bovine testes and hyaluronate lyase from Streptomyces hyalurolyticus were obtained from Sigma-Aldrich (Darmstadt, Germany). Linear RGD was purchased from PSL (Heidelberg, Germany). The Thermo Fisher Scientific Live/Dead Double Staining Kit was purchased from Merck (Darmstadt, Germany).

Normal human dermal fibroblasts (NHDF) with fibroblast basal medium (FGM2) and supplements (FGM2 Supplements) were purchased from Promocell (Heidelberg, Germany).

### 4.1. HA Thiolation

HA thiolation was carried out as described in Shu et al. 2002 [4]. In short, sodium hyaluronate with a molecular weight of 74 kDa was thiolated with 3,3′-dithiobis (propanoic dihydrazide) at the carboxyl group. With different reaction times, different degrees of thiolation were achieved (Table 2) that were subsequently determined with the colorimetric Ellman’s assay [15] in triplicate.

### 4.2. HA Hydrogel Formation and Determination of Free Thiols

HA-DTPH-Ox: HA-DTPH was dissolved at 3% (*w*/*v*) in DPBS (-CaCl_2_, -MgCl_2_) and poured in a petri dish with 3 cm diameter. It was then incubated for 3 days under stirring at 300 rpm. After 3 days of incubation, hydrogels were incubated in 0.3% H_2_O_2_ for 1 h and incubated afterwards for 24 h in MilliQ water.

HA-DTPH-Cl^+^: For preparing HA-DTPH-Cl^+^ hydrogels, an equimolar ratio of ionic cross-linker to remaining negative groups of HA-DTPH was used. HA-DTPH 4% (*w*/*v*) was dissolved in borate buffer (150 mM, pH 8.5) by sonication for 15 min. Ionic crosslinker was also dissolved in borate buffer (150 mM, pH 8.5) in an equimolar ratio to negative groups on the HA-DTPH. The solutions of HA-DTPH and ionic cross-linker are mixed in a 3:7 ratio, yielding a final concentration of 2.8% (*w*/*v*) HA-DTPH. After sonication, the solutions were centrifuged at 1.5 rpm for 2 min and then the ionic cross-linker solution was added to the HA-DTPH solution and mixed by slowly pipetting to get a homogenous mixture. The gelation solution was poured into a cylindrical Teflon-mold and covered with a glass slide. For the gelation of hydrogels with dHA^+^, hydrogels were incubated over night at RT. For gelation of hydrogels with GluA^+^ and NH_4_^+^ the pH is adjusted to 7.4 prior to incubation over night at RT. An overview of used hydrogel systems is given in Table 3.

To measure the number of free thiols in the hydrogels an adapted Ellman’s assay was used. To calculate the percentage of reacted thiols and therefore assess the efficiency of crosslinking measured absorbance was normalized to t_0_.

### 4.3. Mechanical Measurements

For the mechanical characterization of swollen hydrogels, a uniaxial compression test between parallel plates in a NanoBionix Universal Testing System (MTS Systems, Eden Prairie, MN, USA) was performed. The instrument applies an increasing strain, between 0 and 10% and measures the resulting stresses. To obtain the respective Young’s moduli, these data were analyzed in the linear-viscoelastic region between 0 and 5% compression by a linear fit as described in [5]. For an exemplary graph of the linear fit of the stress–strain ratio of HA-DTPH^58%^ without and with the three different ionic cross-linker see Figure A2 of Appendix A.

### 4.4. Swelling Ratio in Water and Varying Ionic Solutions

Swelling studies were performed in ddH_2_O at RT and varying ionic strength solutions 50 mM, 150 mM, 300 mM of NaCl and CaCl_2_. Prepared hydrogels were incubated on a shaker in different solutions (1 mL) until reaching the equilibrium in a 24-well plate. After incubation, the swollen weight was measured and the hydrogel was freeze dried for 48 h. The swelling ratio was determined by weighing the swollen gel (*Gel_s_*) and the dried hydrogel (*Gel_d_*) using Equation (1).
(1)QM=GelsGeld

### 4.5. Enzymatic Degradation

Prepared hydrogels were put into a 24-well plate in solutions of hyaluronic acid-degrading enzymes and PBS as a negative control, where the hydrogels should not have been degraded. All enzymes were used at a concentration of 100 U with a volume of 1 mL for each hydrogel. Hydrogels were incubated with these enzymes at 37 °C and soft shaking 100 rpm, while enzyme and buffer solutions were exchanged every 48 h. For calculating the rate of degradation, the weights of the hydrogels were measured at different time points; after 1 h, 2 h, 4 h, 6 h, 9 h, 12 h, 36 h, 48 h, 72 h, 5 d and 7 d of degradation. The weight measurements at different time points were plotted against weight loss and fitted to an exponential decay phase. From this data, the half-life (t_1/2_) of each hydrogel was determined, corresponding to the time frame in which the hydrogel lost half of its initial weight.

### 4.6. Cell Encapsulation and HA-DTPH-dHA^+^-RGD Preparation

HA-DTPH^58%^ and the dHA^+^ were dissolved in FGM2 and borate buffer, respectively. Afterwards, the cross-linker solution was added to the HA-DTPH^58%^ solution by gently pipetting and the mixture left for gelation for 60–120 min until the viscosity was sufficient to prevent fast-flowing/dripping of the mixture inside the tube. To gain RGD incorporated hydrogels, linear RGD was first incubated for 1 h with the dissolved HA under sterile conditions. The linear RGD peptide was equipped with an acrylamide group, which was introduced into the side chain of the C-terminal lysine. The acrylamide group gives the opportunity to form a covalent bond with the thiol groups performing a Thiol–Michael addition click reaction.

Then, the gelatinous liquid was slowly pipetted onto the previously prepared NHDF cell pellet (FGM2 supernatant was removed carefully) and the pellet was gently re-suspended (using a cut 200 µL tip) to achieve cell dispersion within the entire pre-gelled liquid. Finally, the liquid–cell mixture was pipetted into several provided cut syringes, used as sterile, small sample holders, and left for the remaining gelation reaction in 37 °C CO_2_ incubator covered with parafilm. Additionally, one control hydrogel without cell embedding was prepared. The stable cell-containing hydrogels and the control were then transferred to a 12-well plate the next day and covered with 2 mL FGM2. These hydrogels were stored in the 37 °C CO_2_ incubator for further analysis.

### 4.7. Live/Dead Staining

To analyze the biocompatibility of the hydrogel and guarantee the viability of encapsulated cells within the hydrogel over a broader time range, cell viability was measured after 72 h. Therefore, after 72 h incubation of cells inside the hydrogels, the cell culture medium was removed from the wells and hydrogels were gently washed with PBS once. Next, 200 µL of staining solution was added on top of each hydrogel and incubated for 45 min. Afterwards, staining solution was removed, and the hydrogels were placed on a cover slip and imaged. The staining solution contained cell-permeable dye calcein AM yielding green fluorescence (excitation maximum: 494 nm; emission maximum: 517 nm) to stain live cells, and red fluorescent dye ethidium homodimer-1 (excitation maximum: 517 nm; emission maximum: 617 nm) to stain dead cells.

### 4.8. DAPI/Phalloidin Staining

To determine cell spreading inside of the hydrogel as a response to incorporated RGD, hydrogels were stained with DAPI and Phalloidin 24 h post-encapsulation. As, for the used primary cell line, it is known that cell spreading takes place within 6 h, the time-point was chosen to determine if the response to a binding ligand is diminished in a 3D environment. Cells inside the hydrogel were first incubated with 4% PFA in PBS for 20 min. and subsequently permeabilized with 0.1% Triton X-100 in PBS for 5 min. After the permeabilization procedure, samples were treated with 1% BSA in PBS for 10 min. Subsequently, samples were treated with DAPI (1:1000) and Phalloidin FITC (1:100) and imaged.

### 4.9. Microscopy Imaging

Cells encapsulated inside of the hydrogel for live cell imaging experiments were imaged with a Leica DMi8 inverted fluorescent widefield microscope equipped with an X-Cite 200DC light source (200 W), a sCMOS camera (Leica DFC9000GT, Leica Biosystems, Nussloch, Germany) using 10× objective (HC PL FLUOTAR, NA 0.32, PH1). To generate composites and adjust the brightness contrast of the images, “Fiji” (ImageJ 1.15 h) is used.

## 5. Conclusions

We have demonstrated how physical crosslinks through small charged molecules (NH_4_^+^, GluA^+^, dHA^+^) open up a new biocompatible polymerization reaction for thiolated HA hydrogels with tunable mechanical properties. These systems are ideally suited as 3D cell culture model systems mimicking the ECM, and could be further supplemented with ECM proteins or adhesive peptides. Furthermore, these hydrogels might be of interest for improving drug delivery strategies in vivo or HA supplementation, e.g., in arthritic joints. Additionally, this system underlines the importance of physical interactions within the ECM and the possible role of HA in regulating the storage of small, charged biomolecules within the ECM of various tissues of our body.

Overall, this system has several advantages including: (1) simple chemistry, (2) tunable mechanical stiffness, and (3) biocompatible hydrogel synthesis. The ability to control and tune the mechanical stiffness makes the hydrogel system usable for various biomedical applications. This includes cell study behavior and differentiation studies in a 3D environment.

## Figures and Tables

**Figure 1 molecules-25-04166-f001:**
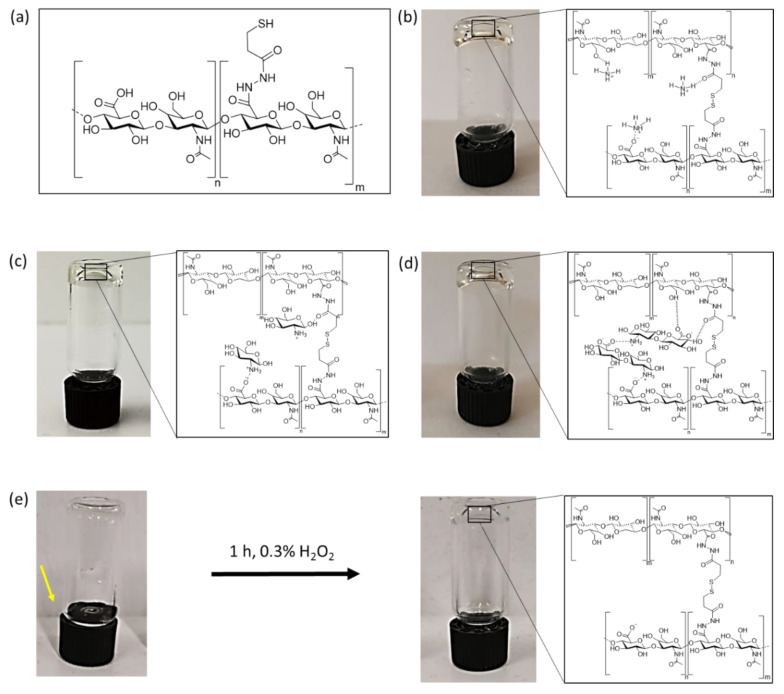
Inverted tube test utilized for hydrogel synthesis with and without ionic cross-linker. (**a**) Chemical structure of thiol functionalized HA; (**b**) thiol functionalized HA (HA-DTPH^29%^) with NH_4_^+^, (**c**) HA-DTPH^29%^ with GluA^+^, (**d**) HA-DTPH^29%^ with dHA^+^ and (**e**) HA-DTPH^29%^ oxidized with H_2_O_2_.

**Figure 2 molecules-25-04166-f002:**
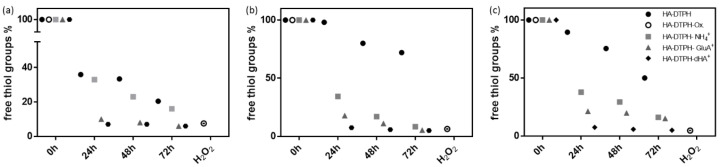
Free thiol group amount within hydrogel network determined by Ellman’s assay. (**a**) for HA-DTPH^29%^, (**b**) HA-DTPH^42%^ and (**c**) HA-DTPH^58%^.

**Figure 3 molecules-25-04166-f003:**
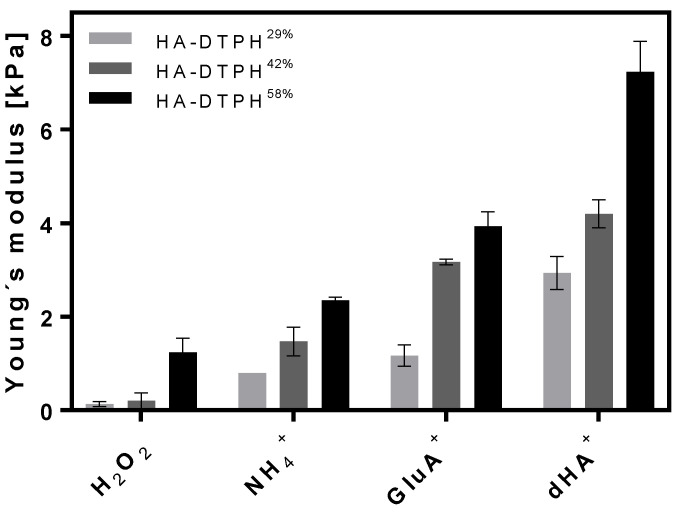
Young’s modulus of hydrogels with three thiolation degrees of HA-DTPH-Ox. and HA-DTPH-Cl^+^ hydrogels.

**Figure 4 molecules-25-04166-f004:**
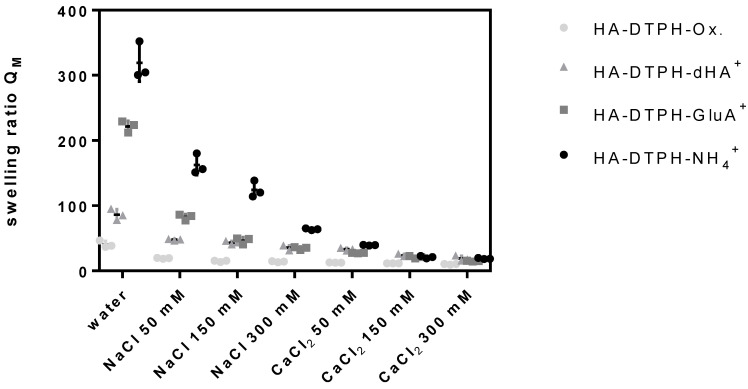
Swelling ratio of HA-DTPH^29%^-Cl^+^ hydrogel and HA-DTPH^29%^-Ox. In water and changing ionic strength and ions.

**Figure 5 molecules-25-04166-f005:**
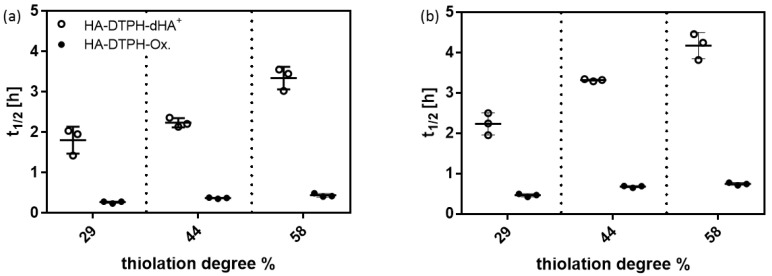
Enzymatic degradation of HA-DTPH-dHA^+^ and HA-DTPH-Ox. (**a**) in 100 U/mL hyaluronate lyase in PBS and (**b**) in 100 U/mL hyaluronidase in PBS.

**Figure 6 molecules-25-04166-f006:**
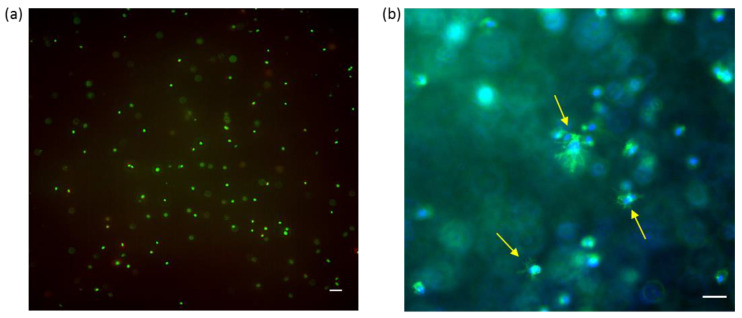
Normal human dermal fibroblasts (NHDF) in HA-DTPH^58%^-dHA^+^ hydrogel. (**a**) Live/Dead staining of NHDF embedded in HA-DTPH^58%^-dHA^+^. Homogenous distribution of NHDF cells inside the hydrogel. Living cell (green fluorescence) and dead cell (red florescence). Image was taken as a Z-stack and max. intensity was performed via ImageJ. Scale bar = 50 µm (**b**) Composite images taken from NHDF embedded in HA-DTPH^58%^-dHA^+^−5% RGD hydrogel. Spreading of fibroblasts (indicated with yellow arrows) was observed 24 h post-embedding inside the hydrogel. Cell nuclei are stained with DAPI. Actin is directly stained with phalloidin. Scale bar = 50 µm.

**Table 1 molecules-25-04166-t001:** Young’s modulus dependency of HA-DTPH^58%^ with varying dHA^+^ equivalents.

Cross-Linker dHA^+^ Equivalents	0.5	1.0	1.5	3.0	5.0
Young’s modulus [kPa]	1.5 ± 0.2	6.7 ± 0.5	9.6 ± 0.6	17.9 ± 2.5	30.1 ± 4.6

**Table 2 molecules-25-04166-t002:** Different degrees of thiolation for 74 kDa HA were achieved with longer reaction times. This table summarizes the different reaction times, in conjugation with the resulting degrees of thiolation for 74 kDa. HA-DTPH with a thiolation degree of 29 ± 4%, 42 ± 2 and 58 ± 5% were used.

Molar Ratio of HA:DTP:EDCl	HA Batch (Lifecore-Biomdical)	Reaction Time	Degree of Thiolation HA-DTPH^x^
1:1:1	024367	20 min	29 ± 4%
1:1:1	024367	40 min	42 ± 2%
1:1:1	024367	60 min	58 ± 5%

**Table 3 molecules-25-04166-t003:** Overview of used hydrogel systems with additionally used ionic cross-linker (including abbreviation for each ionic cross-linker).

*Polymer Backbone*	*Ionic Cross-Linker*	*Hydrogel System*		*Synthesis*
*Thiol-functionalized HA (HA-DTPH)* *with three different thiolation degrees (29%, 42%, 58%)*	-	HA-DTPH-Ox.		via oxidation with H_2_O_2_
	ammonium chloride (NH_4_^+^)	HA-DTPH-Cl^+^ (HA-DTPH + ionic cross-linker)	HA-DTPH-NH_4_^+^	HA-DTPH + ionic cross-linker without oxidation
	charged glucosamine (GluA^+^)		HA-DTPH-GluA^+^	
	synthesized positively charged disaccharide unit of HA (dHA^+^)		HA-DTPH-dHA^+^	

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
