# Peer review of "Small Physical Cross-Linker Facilitates Hyaluronan Hydrogels"

_molecules, 2020, doi:10.3390/molecules25184166_

Round 1

Reviewer 1 Report

It is well known that the hyaluronic acid (HA) exhibits excellent biocompatibility and biodegradability and many important applications in biomedical area. This manuscript reported that the thiol modified HA reacted with different ionic cross-linkers, i.e. NH4+, GluA+ and dHA+, to form hydrogels, which possess tunable swelling ratio, Young’s modulus, enzymatic degradation, and cell encapsulation. Definitely, some results are interesting and attractive. But the following questions should be mentioned and revised in the new version.

i.The authors should analyze the complicated interaction in the hydrogel system and establish the relationship between structure and property. If possible, could the authors measure the zeta-potential of hydrogels to provide some evidence on the electrostatic interaction? Why do the HA-DTPH-Ox hydrogel with low swelling ratio possesses low Young’s modulus?

ii. In abstract and conclusion, please list three ionic cross-linkers directly. Otherwise, the readers confuse a lot.

iii. Please correct the typos, for example, Line 83, “dusilphide bond formation”; Line 220, “swelling ration”. Please check the spelling carefully before submission.

It can be accepted for publication after a revision.

Author Response

Dear Reviewer 1,

thank you for your revision. It is highly appreciated.

  1. The authors should analyze the complicated interaction in the hydrogel system and establish the relationship between structure and property. If possible, could the authors measure the zeta-potential of hydrogels to provide some evidence on the electrostatic interaction? Why do the HA-DTPH-Ox hydrogel with low swelling ratio possesses low Young’s modulus?

Response to point 1: Indeed, a closer analysis of the structure induced by the different additional electrostratic interactions, would be very interesting. Unfortunately, we are not able to perform any zeta potential experiments under the current circumstances. We did include a more in-depth discussion regarding the low swelling ratio of the HA-DTPH-Ox hydrogel: we assume it is the result of low osmotic forces compared to hydrogels prepared with ionic cross-linker (see lines 267-275).

2. In abstract and conclusion, please list three ionic cross-linkers directly. Otherwise, the readers confuse a lot.

Response to point 2: Yes, you are right! We named the cross-linker in the abstract and conclusion and additionally listed the used hydrogel systems with the used ionic cross-linker in Tab.3 in Material & Method.

3. Please correct the typos, for example, Line 83, “dusilphide bond formation”, Line 220, “swelling ration”. Please check the spelling carefully before submission.

Response to point 3: We corrected all the mentioned typos and additionally checked the whole manuscript for typos.

Reviewer 2 Report

Erikci et al. have studied an alternative way of crosslinking hyaluronan (HA) hydrogels via small biological molecules acting as physical crosslinkers and compared it with a more well-known crosslinking method of thiol-groups using hydrogen peroxide. They successfully show hydrogel formation, characterization of relevant properties and biocompatibility. The topic and results are interesting and relevant.

However, there are several shortcomings in the manuscript that could be improved:

1) The introduction is quite short and would benefit from having more references to the usage and production methods of HA-based hydrogels.

2) As the HA thiolation and synthesis of crosslinkers are previously published methods, their previous use should be also explored more in the introduction.

3) The Materials&Methods chapter would benefit from having a table of all the studied hydrogels and their abbreviated names used here.

4) The cell attachment was enhanced by addition of RGD into the hydrogel, but was this peptide only physically entrapped inside the hydrogel network or is there expected to be stronger binding of it? Also, as spreading cells exert forces on the surroundings molecules where they attach, is the free RGD enough to facilitate proper cell spreading?

5) Rheology is briefly mentioned in abstract and in results, but no data is shown and the used equipment and parameters are not listed in Materials&Methods. The rheological analysis should either be shown or left out entirely, now it does not benefit the manuscript.

6) The Results chapter starts with text (lines 55-68) that seems to belong at least partially to the Materials&Methods.

7) The discussion has very little comparison of results from this manuscript to the other manuscripts in the field. 

8) More minor notes on the methodology: The list of reagent providers lists Sigma-Alrich twice, this could be combined (lines 238-242). The used displacement rate is not mentioned under mechanical testing (lines 273-276), even though this is an important parameter for viscoelastic materials such as HA hydrogels. The handling of hydrogels in compression as soft as under 2kPa Young's modulus could be commented, as this is usually difficult. The choice of doing cell testing only with HA-DTPH-dHA+ should be justified (chapter 4.6). Likewise, justification is needed for the choice of performing gelation with cells in a cut syringe first and transferring the ready sample to well plate later, instead of producing the sample directly in a well plate (chapter 4.6). The live/dead staining was done after 72h, but DAPI/Phalloidin staining after just 24h, how were the timepoints chosen (lines 308-319 & 167-173)?

Author Response

Dear Reviewer 2,

thank you for your revision. It is highly appreciated.

  1. The introduction is quite short and would benefit from having more references to the usage and production methods of HA-based hydrogels.

Response to point 1: We added a paragraph about the usage and production methods of HA based hydrogels, with a specific focus on different functionalization and cross-linking methods (lines 33-65).

  1. As the HA thiolation and synthesis of cross-linkers are previously published methods, their previous use should be also explored more in the introduction.

Response to point 2: We described hydrogel system based on thiolated HA in more detail and included possible applications of these systems (also see lines 33-65).  

  1. The Materials&Methods chapter would benefit from having a table of all the studied hydrogels and their abbreviated names used here.

Response to point 3: Yes, you are absolutely right! This might avoid some confusion. We added a table in the Material&Methods (see Tab. 3)

  1. The cell attachment was enhanced by addition of RGD into the hydrogel, but was this peptide only physically entrapped inside the hydrogel network or is there expected to be stronger binding of it? Also, as spreading cells exert forces on the surroundings molecules where they attach, is the free RGD enough to facilitate proper cell spreading?

Response to point 4: Thank for pointing out this issue. We described the employed RGD peptide and cross-linking method in the Material&Methods and in line 216. As the linear RGD was covalently bond to the thiolated HA it facilitated great cell spreading in 3D.

  1. Rheology is briefly mentioned in abstract and in results, but no data is shown and the used equipment and parameters are not listed in Materials&Methods. The rheological analysis should either be shown or left out entirely, now it does not benefit the manuscript.

Response to point 5: We made sure to refer to mechanical measurements only in the manuscript now and added a more precise description of the rheological method used to obtain the Young’s moduli in the Materials&Methods. Regarding this analysis, we added a Figure in the appendix (Figure A2) with exemplary data analysis for HA-DTPH58% without and with the three different cross-linker.

  1. The Results chapter starts with text (lines 55-68) that seems to belong at least partially to the Materials&Methods.

Response to point 6: We apologize, if the start of the results part sounds more like a Materials&Methods description. We shortened this first paragraph significantly to focus on the main message and the details of the employed cross-linker (new lines 90-92).

  1. The discussion has very little comparison of results from this manuscript to the other manuscripts in the field.

Response to point 7: Thank you for pointing this out, we included further references to literature in the discussion now (lines 279-288)

  1. More minor notes on the methodology: The list of reagent providers lists Sigma-Alrich twice, this could be combined (lines 238-242). The used displacement rate is not mentioned under mechanical testing (lines 273-276), even though this is an important parameter for viscoelastic materials such as HA hydrogels. The handling of hydrogels in compression as soft as under 2kPa Young's modulus could be commented, as this is usually difficult. The choice of doing cell testing only with HA-DTPH-dHA should be justified (chapter 4.6).

Response to point 8: We combined the two reagents purchased from Sigma-Aldrich as suggested.

We specified the linear fit of the stress-strain relation employed during mechanical testing now on line 363-367: the data were analyzed in the linear-viscoelastic region between 0 and 5 % compression by a linear fit.

Since, for cell encapsulation studies hydrogel polymerization has to happen in an appropriate time, in order to achieve a homogenous cell distribution within the hydrogel system we tested out which thiolation degree (mainly HA-DTPH58%) and ionic cross-linker (dHA+) gave the most satisfying result. Cell encapsulation is shown exemplary for this hydrogel to demonstrate the general applicability as suitable cell-encapsulation materials.

9. Likewise, justification is needed for the choice of performing gelation with cells in a cut syringe first and transferring the ready sample to well plate later, instead of producing the sample directly in a well plate (chapter 4.6).

Response to point 9: A cut syringe was first of all used to have the flexibility to transfer the polymerized hydrogel onto a glass cover slip. Using the syringe was not just beneficial for the whole handling procedure but we could also save up components.  We added the respective justification in line 402.

10. The live/dead staining was done after 72h, but DAPI/Phalloidin staining after just 24h, how were the timepoints chosen (lines 308-319 & 167-173)?

Response to point 10: The time-points are chosen depending on the read-out. Since biocompatibility testing requires a longer test phase, the viability was chosen after 72h. In contrast, cell spreading of NHDFs celss takes approximately 6 h on cell culture plastic, therefore we observed cell spreading behavior at an earlier time point. We included a short justification in lines 408-409 and lines 418-421.

Round 2

Reviewer 1 Report

The authors have revised the manuscript adequately. It can be accepted for publication on this journal.

Reviewer 2 Report

The revision has significantly improved the manuscript and therefore I recommend accepting it for publication at Molecules.

The only detail missing is compression speed used in mechanical testing, which could still be added in Materials&Methods section.